# Exposure to CuO Nanoparticles Mediates NFκB Activation and Enhances Amyloid Precursor Protein Expression

**DOI:** 10.3390/biomedicines8030045

**Published:** 2020-02-27

**Authors:** Xiaoyang Mou, Alexander Pilozzi, Breeya Tailor, Jing Yi, Catherine Cahill, Jack Rogers, Xudong Huang

**Affiliations:** 1Department of Chemistry and Biochemistry, Rowan University, Glassboro, NJ 08028, USA; xymouwin@gmail.com; 2Neurochemistry Laboratory, Department of Psychiatry, Massachusetts General Hospital and Harvard Medical School, Charlestown, MA 02129, USA

**Keywords:** Alzheimer’s disease, NFκB, IκBα, Aβ, amyloid precursor protein, nanoparticles, engineered nanomaterials, nanoneurotoxicity

## Abstract

Amyloid precursor protein (APP) is directly related to Aβ amyloidosis—a hallmark of Alzheimer’s disease (AD). However, the impact of environmental factors upon APP biology and Aβ amyloid pathology have not been well studied. The increased use of nanoparticles (NPs) or engineered nanomaterials (ENMs) has led to a growing body of evidence suggesting that exposure to metal/metal oxide NPs, such as Fe_2_O_3_, CuO, and ZnO, may contribute to the pathophysiology of neurodegenerative diseases such as AD through neuroinflammation. Our previous studies indicated that exposure to CuO nanoparticles (CuONPs) induce potent in vitro neurotoxicity. Herein, we investigated the effects on APP expression in neuronal cells exposed to different metal oxide NPs. We found a low dose of CuONPs effectively activated the NFκB signaling pathway and increased APP expression. Moreover, the inhibition of p65 expression using siRNA abolished CuONP-mediated APP expression, suggesting that NFκB-regulated APP expression in response to CuONP exposure may be associated with AD pathology.

## 1. Introduction

Alzheimer’s Disease (AD) is a progressive brain disorder, defined by neuronal and synaptic loss in the hippocampus, cerebral cortex, and adjacent regions. Symptoms of the disease consist of memory loss, paranoia, cognitive impairment, confusion, and language deficiencies [1,2,3,4]. AD primarily affects the elderly [5], involving 50% of people over the age of 85 [6], and is the number one cause of dementia [7]. Regrettably, the rising cost of dementia is estimated at $818 billion making it a colossal socioeconomic burden [8]. Early studies of the disease have suggested the onset is associated with the extracellular deposition of abnormal β-amyloid peptides (Aβ) and intracellular accumulation of neurofibrillary tangles (NFT), which are aggregates of hyperphosphorylated tau proteins [6]. In addition to damaged brain tissue and apoptosis [9], clinical investigations have indicated the brains of AD patients exhibit high levels of inflammatory chemokines and cytokines. In addition, there is a notable proliferation of activated microglia [10], a characteristic of excessive inflammatory and innate immune response [11,12,13,14,15,16]. As such, advancements in the field have deemed neuroinflammation a salient pathophysiological feature of AD. However, environmental contributors of the disease are not well understood [17,18].

Indeed, a disruption of homeostasis initiates a cascade of inflammatory cytokines and release of transcription factors [19]. A surfeit of evidence indicates that aging plays a role in increasing oxidative stress, DNA damage, and other imbalances, yet the mechanisms for these are not elucidated [20]. Recently, studies demonstrated processes related to aging and AD pathology by stimulation of the NFκB pathway [21]. For example, expression levels for NFκB were found to increase with age. Moreover, progeria patients, who exhibit features of premature aging, also displayed high levels of NFκB activation. This may be attributed to an increased binding of NFκB/p65DNA. Conversely, attenuating NFκB levels reversed multiple features of aging pathology [21].

Aβ1–42, one of the two forms of Aβ commonly found in amyloid plaques [1], induces an inflammatory response by the activation of the Toll- and NFκB-like signaling pathways in the Drosophila AD model [22]. Persistent NFκB activation in rat primary glial cells was induced by a combination of Aβ1–42 and IL-1β [23]. The genetic deletion of TNFα1R inhibited β-secretase expression and Aβ generation in AD mouse models [24]. The non-steroidal anti-inflammatory drug (NSAID) Flurbiprofen, an NFκB inhibitor, suppressed β-secretase activity and effectively reduced Aβ1–42 levels [25] as well as Aβ amyloid plaque formation in AD transgenic mice [26]. However, it is unknown whether the NFκB pathway directly controls amyloid precursor protein (APP) expression and causes excessive Aβ1–42 deposition in brain tissues. Trace metals are important for cellular functions and enzymatic processes. However, dyshomeostasis is a double-edged sword as deficiencies in copper leads to brain function impairments and the promotion of reactive oxygen species (ROS) [27]. Respectively, insults are mitigated through the upregulation of inflammatory pathways triggering NFκB to produce cytokines such as IL-1, IL-6, and IL-8 [28] and type 1 interferon [29]. Moreover, studies showed that Fe and Cu promoted reactive oxygen species (ROS) and increase neuroinflammation [30]. Furthermore, increased levels of Cu are found with age, leading to exacerbated levels of ROS [31].

Likewise, high levels of biometals are found in the brains of those with AD, including Cu, Fe, and Zn. Abnormal Cu enrichment near/within Aβ amyloid plaques may implicate the particles in AD pathogenesis [32,33,34,35]. For example, the formation of insoluble Cu-Aβ1–42 led to neurotoxicity, tissue damage, and increased brain inflammation. This occurs due to high-affinity interactions between copper ions and Aβ1–42 peptides generating excessive ROS in vitro. Additionally, Cu ions activated the NFκB pathway and induced Cox-2 expression in the livers and lungs of rats [36]. Another study showed biometals binding to Aβ-affected amyloid aggregates [37]. Moreover, analysis of environmental pollutions revealed ambient air contains diverse toxic nanomaterials that exhibit significant negative effects on human health [38,39,40,41,42,43].

Engineered nanomaterials (ENMs) or nanoparticles (NPs) range between 1–100 nm [44] in at least one dimension, with atoms organizing close to the surface [45]. Their structure results in high reactivity due to increased surface areas relative to their volume [46]. Some NPs have been used as biocides [47], yet the properties of benign NPs have been harnessed for drug delivery [48] and industrial use, thus health risk validation is imperative [44]. Studies have shown that ZnONP exposure activated inflammatory responses in A549 cells [49] and interstitial inflammation through lymphocytic infiltration [50]. Moreover, increased migration levels of microphages were found in response to pulmonary inflammation in correlation by Fe_2_O_3_NP exposure in vitro [51]. Furthermore, exposure to 20–40 mg/kg in rats induced exacerbated free radicals in lung tissue [48]. Although not necessarily in the brain, nanoparticles have been well studied and shown to exhibit a wide array of toxic effects on the body.

CuONPs in particular have applications in semiconductors, electronic chips, and heat transfer nanofluids due to their superior thermophysical properties [52]. However, intraperitoneal injections of CuNPs at concentrations between 10–70 nm have been found to result in cognitive impairments [53], blood brain barrier (BBB) damage [54], loss of myelinated fibers, and oxidative damage in the hippocampus of rodents [55]. CuONPs in particular have been found to be detrimental to the function of the brain, inhibiting the action potentials of hippocampal CA1 neurons [56]. The particles have also been reported to exert a wide array of genotoxic effects on brain cells as well [57]. Nonetheless, it remains unclear whether environmental exposure to CuONPs is involved in the AD pathogenesis and if these ENMs can activate the NFκB pathway or affect APP expression in human neuronal cells. 

While it may be unclear how copper nanoparticles contribute to AD, there is compelling evidence that nanoparticles do in fact contribute to AD pathology. Indeed, there certainly seems to be an association, as various metal nanoparticles, including copper, are abnormally abundant in individuals who have Alzheimer’s disease, possibly due to a tendency to promote Aβ aggregation [58,59].Young residents of Mexico City, who are exposed to particulate matter in the excessive air pollution in the area, were found to exhibit AD pathological hallmarks in their brainstems in 99.5% of cases; this included infant specimens [60]. Nonetheless, the exposure to NPs can implicate protective roles from aging, thus understanding the mechanism is critical.

Previously we identified an essential need for a model of cell death due to apoptosis pathways associated with debilitating pathologies versus normal physiology. As such, we presented a fully automated cellular image analysis to quantify the viability of human H4 cells exposed to CuONPs [61]. Then, we tested the cytotoxicity of CuONPs, specifically due to their versatile use in medicine and industrial applications, and found that the viability of human H4 and SH-SY5Y cells was negatively affected. We concluded that additional studies were needed to understand the neurotoxic pathways [42,61].

In our previous work, we investigated the molecular mechanism behind the detrimental effects of CuONP exposure and how it may be linked to AD pathology [43]. Herein, to determine the role of NFκB in the regulation of APP gene expression in response to CuONP exposure in human neuronal cells, we utilized the RNAi technique to knockout the expression of the NFκB p65 subunit to characterize its effects on APP gene expression. Lastly, we tested the effects of CuONPs upon NFκB activation and APP expression. Our data indicated that NFκB regulates APP expression in response to CuONP exposure, and CuONP-mediated NFκB activation may be associated with AD pathogenesis.

## 2. Experimental Section

### 2.1. Cell Culture

Human neuroblastoma SH-SY5Y and neuroglioma H4 cells (ATCC, Manassas, VA, USA), and rat pheochromocytoma PC12 cells (ATCC, Manassas, VA, USA) were cultured in Dulbecco’s modified Eagle medium (DMEM) (Invitrogen, Carlsbad, CA, USA) containing 10% fetal calf serum (Invitrogen, Carlsbad, CA, USA) and 1% penicillin-streptomycin (GIBCO, Dublin, Ireland) at 37 °C in an incubator (95% humidity, 95% air, 5% CO_2_) for 48 h.

### 2.2. Transfection

SH-SY5Y cells were first transfected with NFκB reporter plasmid (pNFκB-Luc-neo) using the Nucleofactor kit V (AMAXA Biosystems, Cologne, Germany). Then SH-SY5Y cells were divided into two groups and further transfected with either siRNA (control) or p65 siRNA (15 nM, Santa Cruz Biotechnology, Dallas, TX, USA).

### 2.3. Luciferase Reporter Assay

SH-SY5Y cells (2 × 10^3^/well) expressing an NFκB reporter plasmid were plated into a 96 well in triplicate. Cells were exposed to CuONPs (<50 nm in diameter, 29 m^2^/g in surface area, Sigma-Aldrich, St. Louis, MO, USA), Fe_2_O_3_NPs (<50 nm in diameter, 50–245 m^2^/g in surface area, Sigma-Aldrich, St. Louis, MO, USA), and ZnONPs (<100 nm in diameter, 10–25 m^2^/g in surface area, Sigma-Aldrich, St. Louis, MO, USA) in the concentration range of 0–100 μM and incubated for 6 h (95% air, 5% CO_2_, 95% humidity). Luciferase reagent (Caliper Lifescience, Waltham, MA, USA) was added (30 μg/200 μL each well), and after 30 min of incubation luciferase activity was measured by a Veritas™ microplate luminometer (Turner BioSystems, Cambridge, MA, USA).

### 2.4. Western Blot

#### 2.4.1. Nuclear and Cytoplasmic Protein Extraction

Naïve SH-SY5Y cells or SH-SY5Y cells transfected with either control siRNA or p65 siRNA were plated at 2 × 10^6^ cells/well (6 well plate) and exposed to CuONPs (<50 nm in diameter, 29 m^2^/g in surface area, Sigma-Aldrich, St. Louis, MO, USA) in the presence or absence of a potent inhibitor of NFκB activation—ammonium pyrrolidine dithiocarbamate (PDTC) at times between 0–60 min. Cells were lysed, and the nuclear protein were extracted using NE-PER Nuclear and Cytoplasmic Extraction Reagent (PIERCE, Rockford, IL, USA). The total protein levels were determined by a bicinchoninic acid (BCA) protein assay kit (Thermo-Fisher, Waltham, MA, USA) and a Spectra Max M5e plate reader (Molecular Devices, San Jose, CA, USA). Equal amounts of protein per sample were added to NuPAGE loading buffer (Invitrogen, Carlsbad, CA, USA), and boiled for 10 min. Samples were electrophoresed on NuPAGE^®^ 4–12% Bis Tris gels with NuPAGE MES SDS running buffer and transferred to nitrocellulose membranes (Invitrogen, Carlsbad, CA, USA).

#### 2.4.2. Nitrocellulose Membranes

The nitrocellulose membranes were blocked in Tris Buffered Saline with Tween (TBST) (5% non-fat dry milk) for 1 h at room temperature or overnight at 4 °C. Membranes were washed 3 times for 10 min each with 15 mL TBST followed by incubations with either anti-NFκB antibody (1:400, Santa Cruz Biotechnology, Dallas, TX, USA), anti-APP C-Terminal antibody (1:1000, SIGMA-Aldrich, St. Louis, MO, USA), or anti-IκBα antibodies (Santa Cruz Biotechnology, Dallas, TX, USA) overnight at 4 °C. Anti-β-Actin antibody (1:5000, SIGMA-Aldrich, St. Louis, MO, USA) was used to monitor the protein loading. Following the antibody incubations, blots were washed 3 times for 10 min in TBST. After 1 h incubation, this was followed with a secondary antibody—either anti-mouse Ig HRP antibody (1:10,000, GE Healthcare, Chicago, IL, USA) or anti-Rabbit IgG HRP (1:10,000, GE Healthcare, Chicago, IL, USA). Substrate was then added, after 3 times TBST washing, and membranes were visualized on a Versa Doc Imaging System (Bio-Rad, Hercules, CA, USA).

## 3. Results

We analyzed the activation properties of three different metal oxide NPs (CuO, Fe_2_O_3_, and ZnO) on the NFκB luciferase reporter gene in transfected SH-SY5Y cells. Similar to our previous findings, CuONPs affected cell viability [42] and activated the NFκB reporter 7-fold at a dose of 10 µM. This is higher than the 2- to3-fold activation by Fe_2_O_3_NPs or ZnONPs (Figure 1). Notably, the observed fold change decreased substantially as the CuONP concentration increased from 10μM to 100 μM; perhaps this is due to the increased disruption from the toxic effects of 100 μM CuONPs, as the particles’ toxic and genotoxic effects tend to increase with concentration [57,62]. Our previous study found that CuONPs were highly toxic at concentrations of 100 μM [43]. Other studies have also confirmed that any imbalance of some metal homeostasis induced adverse effects to brain pathology [63,64,65].

We further explored the effects of CuONPs on IκBα protein levels on SH-SY5Y cells exposed to CuONPs (10 µM) in the presence or absence of pyrrolidine dithiocarbamate (PDTC), an NFκB activation inhibitor (50 nM), and found decreased levels of IκBα protein at 30 min with the lowest at 60 min (Figure 2A). PDTC inhibited the degradation of IκBα in response to CuONP treatment (Figure 2B,C) at 60 min.

The influence of CuONPs on APP expression was analyzed by administering treatment for 72 h on SH-SY5Y cells. The levels of APP expression were increased up to 5-fold during 6 to 24 h time periods, declining thereafter with a dose level of 10 µM (Figure 3A). No change in the protein concentration of β-actin indicated specificity. The effects of the CuONP dose (0.01–100 µM) on APP protein expression after 6 h of exposure in SH-SY5Y and PC12 cells were detected. TNFα (50 ng/mL) was used as a control for APP induction, and higher levels (10–100 µM) of CuONPs resulted in increased APP expression in SH-SY5Y and PC12 cells (Figure 3B). CuONPs of 10 µM increased APP in both cell types (SH-SY5Y and PC12). However, a 4-fold induction of APP in SH-SY5Y cells compared to 2-fold induction in PC12 cells was found. Notably, this induced change in APP expression was not as large as that of NFκB. A study by Lv et al. showed the binding of Cu ions induced structural change in the amyloid dimer, which can induce AD pathology [66].

Moreover, we explored whether the NFκB pathway was involved in increased APP expression in response to CuONP treatment. The effects of CuONPs on nuclear p65 accumulation and cytoplasmic APP expression were detected in response to the incubation of either TNFα—an NFκB activator and inducer of APP expression—as a positive control (Figure 4A) or CuONPs (Figure 4B) in the presence or absence of PDTC (50 nM) (Figure 4). Lysates were blotted for the presence of p65, APP, and β-actin (control). TNFα of 50 ng/mL (Figure 4A) and CuONPs of 10 µM (Figure 4B) induced the nuclear accumulation of p65 and increased APP expression at 6 h. Additionally, PDTC partially inhibited p65 nuclear accumulation and APP expression in response to TNFα (Figure 4A) and CuONP exposure (Figure 4B).

We conducted a p65 protein knockout test to confirm the involvement in APP regulation in response to CuONP exposure via p65 SiRNA or a siRNA control. The control lysates contained inducible p65 nuclear accumulation in response to CuONP and TNFα treatments, unlike cells transfected with p65 siRNA (Figure 5A). Furthermore, β-actin levels were unaffected by the treatment. APP induction in response to CuONP or TNFα exposure was unaffected by the control siRNA. However, the transfection of p65 siRNA resulted in a significant reduction in basal and inducible APP expression in response to CuONP and TNFα treatments, thus confirming the role of NFκB p65 induction in response to CuONPs (Figure 5B).

## 4. Discussion

The interaction of Aβ with transition metals such as copper, zinc, and iron results in the aggregation of Aβ and the accumulation of reactive oxygen species (ROS), both of which promote AD pathogenesis. Indeed, metals are often found associated with amyloid plaques [37]. This is based on in vitro studies accelerating the aggregation and precipitation into plaques of Aβ, ultimately leading to synaptic dysfunction and accelerated amyloidogenesis [67,68].

Our current findings, that CuONPs are the more toxic than ZnONPs and Fe_2_O_3_NPs, confirm our previous study [42,61]. As indicated in Figure 1, CuONPs induced the most NFκB activation when concentrations of NPs were above 1 μM. This is because dissolute Cu ions from CuONPs are more redox active than Fe ions from Fe_2_O_3_NPs, and CuONPs also have smaller surface areas than Fe_2_O_3_NPs while dissolute Zn ions from ZnONPs are redox inert. Further analysis of the CuONP effects on NFκB activation in brain cells resulting from attenuated cell viability can produce insight into whether attenuation of NFκB activation may reduce AD pathogenesis [69]. Aβ levels were attenuated by suppressing BACE1 through NFκB deactivation, confirming the role of NFκB in Aβ formation [70]. Thus, modulating the NFκB-mediated neuroinflammatory pathway may be a valid therapeutic approach for AD pathology [71,72,73,74].

Experimental data have shown detrimental effects of NP exposure on brain health, which may lead to etiopathogenesis of AD [75], as exposure to ENMs continues to rise in both industrial and consumer applications [43]. There is overwhelming evidence, in the form of preclinical trials, in vitro and in vivo, which support the roles of metals and metal oxides alike in the pathogenesis of AD [43,76].

Herein, our data supports the argument that environmental exposure to ENMs is injurious to our health. As shown in our controlled in vitro assays, CuONP treatment activated NFκB more than ZnONPs and Fe_2_O_3_NPs in pNFκB-Lu-neo SH-SY5Y cell lines. Other studies have concluded that genetic manipulation of the NFκB pathway led to exhaustive inflammatory action, had serious consequences on aging as found in intracellular negative regulators, and led to toxic accumulations of antimicrobial peptides in flies [77]. Thus, it is critical we evaluate the relationships and toxic effects influenced by various NPs, as no index based on biological response is currently available [41]. It is evident that exposure to ambient particulate matter is a significant risk factor for cognitive decline and dementia [78]. Studies such as ours, which elucidate the mechanisms behind particulate toxicity, are vital to understanding how and where the problem needs to be addressed.

We have demonstrated that CuONPs at 10 µM activated NFĸB after 6 h of treatment and enhanced APP expression from 6 to 72 h. At 24 h of treatment, the expression of APP was highest, while APP expression levels decreased at 72 h. Furthermore, APP increased as NFĸB increased, indicating NFĸB-regulated APP expression.

NFκB is made up of multiple subunits including IκBα, p65, and p50. Activation occurs by proteasome degradation of the inhibitor IκB, followed by entry of p65 and p50 into the nucleus. Our study indicated that exposure to CuONPs caused the degradation of IĸBα starting between 30 and 60 min, and later activated NFκB. When NFĸB was over expressed, APP levels were also high. On the other hand, when NFκB was inhibited with PDTC, APP expression was inhibited as well in SH-SY5Y and PC12 cells. This is critical in unearthing the role of inflammation and potential pathways to therapies, given the relevance of APP, inflammation, and the interplay between the two with regard to AD [1,10,12]. Overall, we found APP expression was significantly reduced when NFκB was inhibited, suggesting that NFκB influences APP levels. Overall, our data suggests that CuONP exposure induced NFκB-mediated neuroinflammation and increased AD risk. Thus, as current research efforts have failed to yield effective AD treatment so far, further understanding of the molecular mechanisms related to environmental exposure to CuONPs and its potential contribution to AD etiopathogenesis may offer an alternative perspective.

## 5. Conclusions

The effects of environmental factors, such as exposure to ENMs, on neurodegeneration are not well understood. Thus, understanding neurodegenerative mechanisms relating to exposure to ENMs, such as CuONPs, in neurological diseases such as AD can provide rationale for the regulation of these materials. In this study, we have determined that low-dose CuONP exposure has a direct role in activating the NFκB signaling pathway and increasing APP expression. Moreover, the inhibition of p65 expression using siRNA abolished CuONP-mediated APP expression, suggesting that NFκB-regulated APP expression in response to CuONP exposure may be associated with AD pathology. Hence, regulating environmental exposure to CuONPs may be necessary to lower AD risk and promote overall healthy brain aging.

## Figures and Tables

**Figure 1 biomedicines-08-00045-f001:**
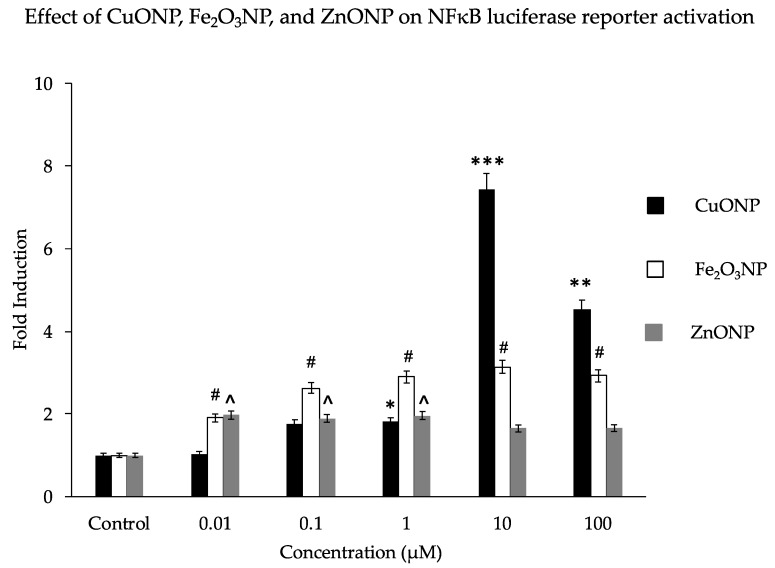
Effect of nanoparticles on NFκB luciferase reporter activation. Human SH-SY5Y cells (2 × 10^4^/well) expressing a NFκB reporter gene (transfected plasmid—pNFκB-Lu-neo) were exposed to concentrations of nanoparticles (NPs) of CuO, Fe_2_O_3_, and ZnO at a range of 0 to 100 μM/mL for 6 h. Luciferase reagent was added and luciferase activation signals were measured using a Veritas™ microplate luminometer. Results are expressed as fold induction. * *p* < 0.01, ** *p* < 0.001, *** *p* < 0.0001, for CuONPs (10 μM, 100 μM) compared to control, untreated, ^#^
*p* < 0.01 for Fe_2_O_3_NPs compared to control, ^ *p* < 0.01 for ZnONPs compared to control.

**Figure 2 biomedicines-08-00045-f002:**
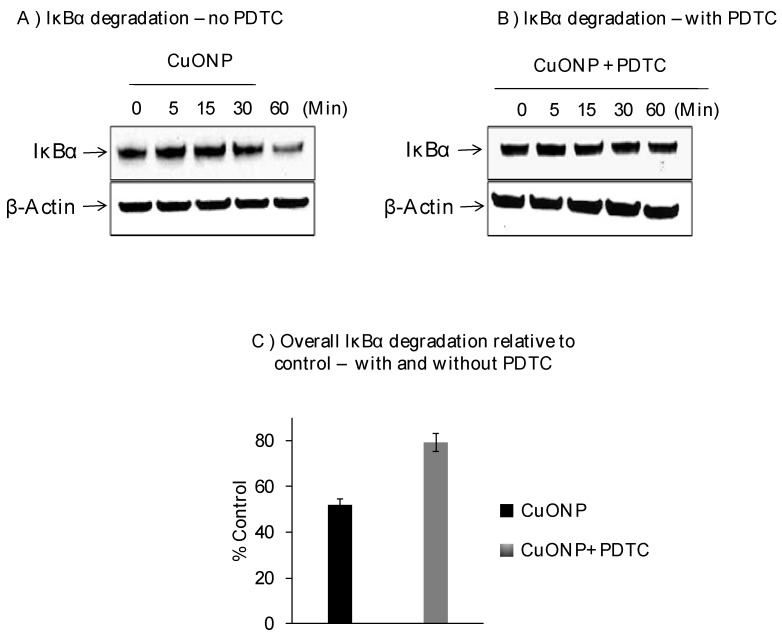
Effect of CuONPs on IĸB-α degradation. SH-SY5Y cells were plated at 2 × 10^6^ cells/well (6 well plate) and exposed to CuONPs (10 μM) in the presence or absence of the potent NFκB inhibitor—pyrrolidine dithiocarbamate (PDTC, 50 nM) at the indicated time points. Cells were lysed, and lysates were Western blotted for the presence of IκB-α—a protein inhibitor for NFκB activation. Blots were collected, digitized, and quantified using a Bio-Rad VersaDoc™ Digital Imaging System (MP4000). Experiments were performed at *n* = 3 independent trials and representative Western blots were presented. (**A**) Western blot from cells exposed to CuONPs but not PDTC; (**B**) western blot from cells exposed to CuONPs and PDTC; (**C**) summary graph of relative degradation (compared to controls) in cells exposed to CuONPs and CuONPs and PDTC.

**Figure 3 biomedicines-08-00045-f003:**
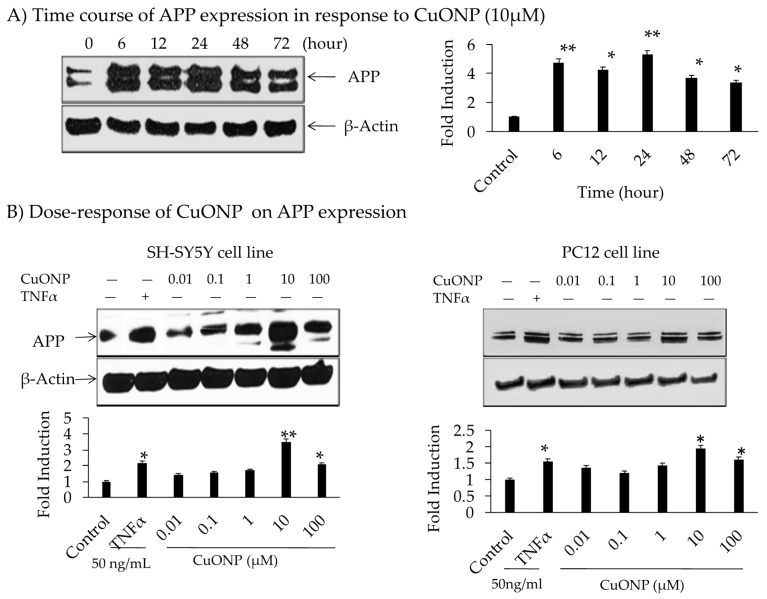
Effect of CuONPs on amyloid precursor protein (APP) expression. (**A**) Time course of APP expression in response to CuONPs (10 µM); (**B**) dose 0–100 µM, response of CuONPs on APP expression in SH-SY5Y cells (left) and PC12 cells (right). In (**A**), cells were harvested at the indicated time points and blotted for APP and β-actin. In (**B**), doses of CuONPs in the range (0–100 µM) and TNFα (50 ng /mL) were used and cells harvested at 6 h and lysates blotted for APP and β-actin. Densitometric analysis from 3 independent experiments were plotted and Student’s t-test were performed to determine levels of significance, * *p* < 0.05, ** *p* < 0.01 CuONPs or TNFα treated compared to control.

**Figure 4 biomedicines-08-00045-f004:**
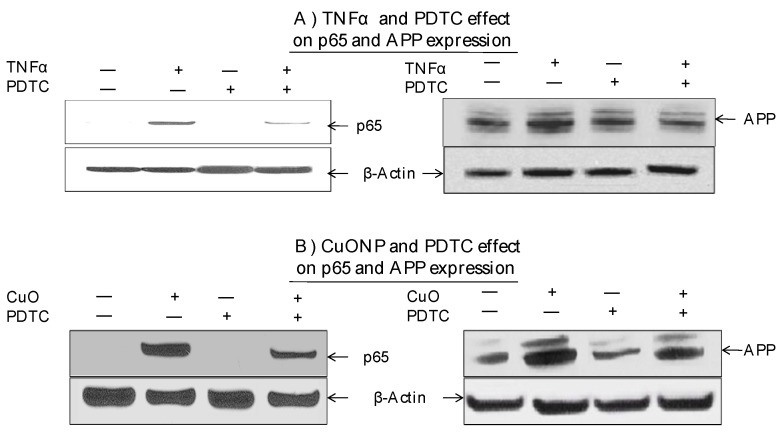
Effect of PDTC on APP and p65 induction by CuONPs. SH-SY5Y cells were plated at 2 × 10^6^ cells/well (6 well plate) and exposed to (**A**) TNFα (50 ng/mL) or (**B**) CuONPs (10 μM) in the presence or absence of PDTC (50 nM) for 6 h. Cells were harvested and lysates (nuclear and cytoplasmic) blotted for p65, APP, and β-actin by Western blotting.

**Figure 5 biomedicines-08-00045-f005:**
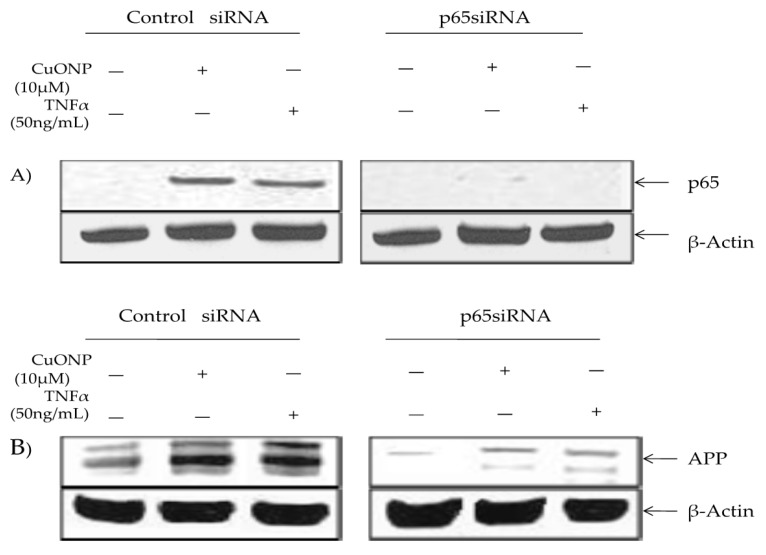
Effect of p65 knockdown on amyloid precursor protein (APP) induction by CuONPs and TNFα. Effects of p65 siRNA (15 nM) on APP induction by CuONPs and TNFα were assessed. SH-SY5Y cells were plated at 2 × 10 ^6^ /well and transfected with either control siRNA or p65 siRNA. After 24 h, transfected cells were exposed to either CuONPs (10 μg/mL) or TNF-α (50 ng/mL) for 6 h. Cells were lysed and lysates blotted for p65 (**A**) or APP (**B**), and β-actin as a control.

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
