# Peer review of "Exposure to CuO Nanoparticles Mediates NFκB Activation and Enhances Amyloid Precursor Protein Expression"

_biomedicines, 2020, doi:10.3390/biomedicines8030045_

Round 1
Reviewer 1 Report
In this manuscript, the authors investigated the effect of different nanoparticles on APP expression and NFκB activation. They showed that using low concentration of CuONP (10 uM) could increase the APP expression in neuronal cells. They suggested that APP expression could be regulated by CuOPN-mediated NFκB activation. This is interesting study that could enhance our understanding of nanoparticles implication in AD pathogenesis. This manuscript is well written with a good standard of English language. The work is worthy of consideration for publication though a few issues/questions need to be addressed/answered first, as described below:
- My main concern is that the authors did not specify the particles size of the NPs. It has been showed that nanoparticles with different size show different effects. In this study the authors did not specify the particle size, particle size should be specified alongside the concentration.
- At higher concentration of CuONP (100 uM), why there is less activation compared to lower concentration, could authors explain the reason?
- It is well known that copper ions are involved in oxidative stress in AD, have the authors checked whether CuONP caused any oxidative damage in cells?
Typing error:
- Page 1, line 18, (to suggest exposure) should be (to suggest that exposure)
- Page 1, line 20, in vitro should be Italic
- Page 1, line 38, (clinical investigations indicated brains) should be (clinical investigations indicated that brains)
- Page 2, line 44, (A surfeit of evidence indicates aging) should be (A surfeit of evidence indicates that aging)
- Page 2, line 71, in vitro should be Italic
- Page 2, line 83, in vitro should be Italic
- Page 3, line 88, CuNP should be CuONP
- Page 3, line 83-88, intraperitoneal injections varying between 10 nm -70 nm of CuNP, this sentence needs rewording.
- Page3, line 109-110, (Herein, we investigated the molecular mechanism behind the detrimental effects of CuONP exposure and how it may be linked to AD pathology [41]),in this sentence the authors are talking about their current study, why they are referring for other people work, why they added reference number 41? This is confusing.
- Page 4, line 148, reformate 4oC.
- Page 10, line 253, at after 6 h treatment, remove (at)
Author Response
Reviewer 1’s Comments and Responses:
In this manuscript, the authors investigated the effect of different nanoparticles on APP expression and NFκB activation. They showed that using low concentration of CuONP (10 uM) could increase the APP expression in neuronal cells. They suggested that APP expression could be regulated by CuOPN-mediated NFκB activation. This is interesting study that could enhance our understanding of nanoparticles implication in AD pathogenesis. This manuscript is well written with a good standard of English language. The work is worthy of consideration for publication though a few issues/questions need to be addressed/answered first, as described below:
- My main concern is that the authors did not specify the particles size of the NPs. It has been showed that nanoparticles with different size show different effects. In this study the authors did not specify the particle size, particle size should be specified alongside the concentration.”
Response: We have specified the sizes of each particle type in the experimental section.
- At higher concentration of CuONP (100 uM), why there is less activation compared to lower concentration, could authors explain the reason?”
Response: Higher concentration of CuONP (100 uM) has caused tremendous cell death via apoptosis (see our previous work in references 42,43,61). This may lessen total NFκB protein levels.
- It is well known that copper ions are involved in oxidative stress in AD, have the authors checked whether CuONP caused any oxidative damage in cells?”
Response: The cellular oxidative damage has been confirmed in other studies, including one of our own; we found a decrease in intracellular thiol levels following CuONP exposure (Shi et al., reference 43).
Typing error:
- Page 1, line 18, (to suggest exposure) should be (to suggest that exposure)”
Response: We have reworded this sentence.
- Page 1, line 20, in vitro should be Italic”
Response: We have italicized all instances of “in vitro”, thank you for pointing this out.
- Page 1, line 38, (clinical investigations indicated brains) should be (clinical investigations indicated that brains)”
Response: We have corrected this error.
- Page 2, line 44, (A surfeit of evidence indicates aging) should be (A surfeit of evidence indicates that aging)”
Response: We have corrected this error.
- Page 2, line 71, in vitro should be Italic”
Response: We have italicized all instances of “in vitro”.
- Page 2, line 83, in vitro should be Italic”
Response: We have italicized all instances of “in vitro”.
- Page 3, line 88, CuNP should be CuONP”
Response: CuNP is correct in this case, as we are referring to a separate study involving copper nanoparticles, not copper-oxide nanoparticles.
- Page 3, line 83-88, intraperitoneal injections varying between 10 nm -70 nm of CuNP, this sentence needs rewording.”
Response: We have changed the wording of this sentence.
- Page3, line 109-110, (Herein, we investigated the molecular mechanism behind the detrimental effects of CuONP exposure and how it may be linked to AD pathology [41]),in this sentence the authors are talking about their current study, why they are referring for other people work, why they added reference number 41? This is confusing.”
Response: We apologize for the confusion; we were referring a previous study, so the use of “herein” was incorrect. We have corrected this error.
- Page 4, line 148, reformate 4oC.
Response: We have corrected this error.
- Page 10, line 253, at after 6 h treatment, remove (at)”
Response: We have corrected this error.
Reviewer 2 Report
Mou et al. report the effect of CuO containing nanoparticles (CuONP) on APP expression in neuronal cells. They founded that CuONP increase APP expression through the activation of NFkB signaling pathway. The authors showed that that NFkB may regulate APP expression in response to CuONP exposure. Importantly, the study has a high relevance to AD pathology.
Main limitations of the study are:
a) There is no any information about metal NPs used in this study. It would be important to know the amounts of different metal oxides per NP. Any other comparison between the content of different oxides in the study will be also relevant. It will allow to compare the effect of NPs containing the same amount of metal oxides, but not the same concentration of NPs. In addition, other characteristic parameters for NP have to be included.
- b) Fig 3a shows transient APP expression in response to CuO NPs peaking at 6-24h. Please, find out when the effect of CuONP will be completely neglected. I think that this information may add knowledge to pathology of AD.
- c) Fig 3b: it looks that CuONPs do not show dose response on APP expression in contrast to Fig.1 where the effect of CuOH on NFkB activation is clearly seen. Please, compare the data of Fig.1 and Fig 3b and discuss it.
- d) Since 10 uM of CuONP is high concentration of NPs, the cumulative effect of small concentrations of CuONP on NFkB activation and APP expression should be also checked.
e) Figure titles are not clear and have to be reworked. For example, Fig.1 “Nanoparticle CuO; Fe2O3, ZnO effect on luciferase reporter activation” has to be corrected. This comment is relevant to Fig 2c and Fig.4. - f) The draft requires additional English editing. For example,
- the numbers in lines 135,148,171 have to be corrected
- the sentence in lines 156-157 requires editing.
- line 196: replace “upper” and “lower” to the left and right, respectively.
Author Response
Reviewer 2’s comments and responses:
Mou et al. report the effect of CuO containing nanoparticles (CuONP) on APP expression in neuronal cells. They founded that CuONP increase APP expression through the activation of NFkB signaling pathway. The authors showed that that NFkB may regulate APP expression in response to CuONP exposure. Importantly, the study has a high relevance to AD pathology.
Main limitations of the study are:
“a) There is no any information about metal NPs used in this study. It would be important to know the amounts of different metal oxides per NP. Any other comparison between the content of different oxides in the study will be also relevant. It will allow to compare the effect of NPs containing the same amount of metal oxides, but not the same concentration of NPs. In addition, other characteristic parameters for NP have to be included.”
Response: We have added the size and surface-area specifications for the different nanoparticles to the experimental section.
“b) Fig 3a shows transient APP expression in response to CuO NPs peaking at 6-24h. Please, find out when the effect of CuONP will be completely neglected. I think that this information may add knowledge to pathology of AD.”
Response: We agree that establishing this would be worthwhile, however it is beyond the scope of the current manuscript. Further experimentation along these lines will likely be performed and reported in a future publication. Thank you for the suggestion.
“c) Fig 3b: it looks that CuONPs do not show dose response on APP expression in contrast to Fig.1 where the effect of CuOH on NFkB activation is clearly seen. Please, compare the data of Fig.1 and Fig 3b and discuss it.”
Response: We note that a dose-dependent response is observed (see the graphs below each western blot in figure 3B), with notably significant fold-changes occurring only at 10µM and, to a lesser extent, 100 µM of CuONP. We have made a note that the magnitude of this change is small compared to that of NFκB.
“d) Since 10 uM of CuONP is high concentration of NPs, the cumulative effect of small concentrations of CuONP on NFkB activation and APP expression should be also checked.”
Response: We agree that determining the effects of consistent exposure to small quantities of CuONPs would be very useful in furthering our understanding the effects of environmental exposure to the particles. However, that is beyond the scope of the current manuscript, but may be investigated and reported in a later publication. Thank you for the suggestion.
“e) Figure titles are not clear and have to be reworked. For example, Fig.1 “Nanoparticle CuO; Fe2O3, ZnO effect on luciferase reporter activation” has to be corrected. This comment is relevant to Fig 2c and Fig.4.”
Response: Figure/figure section titles have been altered; The changes you requested have all been made.
“f) The draft requires additional English editing. For example,
- the numbers in lines 135,148,171 have to be corrected
- the sentence in lines 156-157 requires editing.
- line 196: replace “upper” and “lower” to the left and right, respectively.”
Response: We have corrected these errors, among others. Thank you for pointing them out.
Round 2
Reviewer 2 Report
I am satisfied with the most of revisions made on the manuscript.
However, the authors response:
"Response: We have added the size and surface-area specifications for the different nanoparticles to the experimental section"- was not included in the draft. Please, introduce it to the text and include the discussions correlated the neurotoxic properties of NP via their different characteristics including metal oxides.
Author Response
Reviewer 2’s further comments and responses:
“I am satisfied with the most of revisions made on the manuscript.
However, the authors response:
"Response: We have added the size and surface-area specifications for the different nanoparticles to the experimental section"- was not included in the draft. Please, introduce it to the text and include the discussions correlated the neurotoxic properties of NP via their different characteristics including metal oxides.”
Response: We Thanks for this reviewer’s positive comments upon our manuscript. Between lines 210-212 and highlighted with red color, we added the size and surface-area specifications for the different nanoparticle. In addition, between lines 340-343, we have included discussions correlated the NFκB activation properties of NPs via their different redox activities of relevant dissolute metal ions from metal oxide NPs and surface areas of NPs.